# Multi-Modal Spiking Neural Network for Efficient and Robust Underwater Object Detection

## Abstract

Multi-modal artificial neural networks (ANNs) have demonstrated strong performance gains in object detection by leveraging complementary information from diverse data modalities. However, these gains often come at the cost of substantial increased computational demands due to dense operations and multi-branch architectures. To address these challenges, we propose MMSNN, a novel Multi-Modal Spiking Neural Network for efficient underwater object detection. MMSNN integrates RGB features with Local Binary Pattern (LBP) representation, capturing both fine-grained visual details and illumination-robust texture cues within a spike-driven architecture. At the core of MMSNN is the Spike-Driven Multi-Modal Fusion (SMMF) module, a lightweight yet expressive component designed to enable efficient cross-modal feature interaction. The SMMF uses channel grouping and shuffling to promote localized feature interaction and enhance representational diversity, while its spike-driven attention mechanism reduces computational overhead without compromising discriminative power. Extensive experiments on the RUOD and DUO underwater datasets demonstrate that MMSNN achieves state-of-the-art performance with an excellent balance between robust accuracy and computational efficiency.

## 1 Introduction

Artificial neural network (ANN)-based multi-modal frameworks have been widely adopted in generic object detection (Cao et al. (2023); Li et al. (2025)); however, their use in underwater object detection remains limited, primarily due to high computational demands and the scarcity of publicly available multi-modal datasets. Nonetheless, multi-modal frameworks hold significant promise for underwater object detection, as fusing diverse modalities enables the capture of complementary information and improves robustness under challenging environmental conditions.

To date, only a few studies (Yu et al. (2025); Chen et al. (2024a)) have explored multi-modal detection frameworks that fuse acoustic and optical images to enhance underwater object detection performance. Nevertheless, the absence of publicly available multi-modal underwater datasets significantly hinders further progress in this area. In contrast, generic object detection has benefited from the availability of large-scale multi-modal datasets, such as FLIR (Zhang et al. (2020a)) and LLVIP (Jia et al. (2021)), which has driven the rapid development of multi-modal frameworks. Many studies (Chen et al. (2024b); Li et al. (2025)) employ backbone architectures with parallel branches to extract features from infrared and RGB images, thereby enhancing detection performance. Moreover, several works (Guan et al. (2019); Li et al. (2019)) incorporate illumination-aware modules that enable the detection model to dynamically adjust the weighting of different input modalities based on light conditions, further improving the model's robustness to illumination variations.

Even though multi-modal frameworks show potentials for boosting performance, they still face two main challenges: (1) decreased computational efficiency due to their reliance on multi-branch backbone architectures; and (2) dependence on external data modalities, which require specialized hardware, such as infrared cameras, that can be costly. In this work, we make the following **three contributions** to address these limitations: **First**, we propose a Multi-Modal Spiking Neural Network (MMSNN) for underwater object detection, leveraging the event-driven nature and sparse computa-

tion of SNNs to enhance computational efficiency. **Second**, we utilize Local Binary Pattern (LBP) features as a secondary modality, extracted directly from RGB images, thereby eliminating the need for additional hardwares. LBP features encode local texture and edge patterns that are inherently robust to illumination variations, thereby contributing to the model's robustness. **Third**, we introduce a spike-driven multi-modal fusion module that is fully compatible with the SNN framework. This module is designed to enhance cross-modal information exchange, minimize feature redundancy, and encourage feature diversity.

## 2 RELATED WORK

**Underwater Object Detection.** Deep-learning research on underwater object detection (UOD) has evolved along two main tracks. Early studies (Li et al. (2017); Chen & Fan (2020)) adapted generic two-stage detectors such as Faster R-CNN (Ren et al. (2016)) and R-FCN (Dai et al. (2016)), then shifted to real-time one-stage models like SSD (Liu et al. (2016)) and YOLO (Redmon et al. (2016)). While these adaptions improved inference speed, they remained vulnerable to underwater image degradations. More recent efforts have focused on developing underwater-specific one-stage architectures. These include SSD variants enhanced with multi-scale feature fusion (Pan et al. (2021)), attention mechanisms (Zhang et al. (2020b)) and integrated image enhancement (Zhang et al. (2020c)); YOLO derivatives augmented with domain transfer strategies (Liu et al. (2020)) and image enhancement techniques (Alla et al. (2022)); and, most recently, Swin Transformer (Liu et al. (2024)) and hybrid Transformer (Chen et al. (2023)) architectures have emerged, combining global self-attention with CNN features to better handle blur, color cast and small object detection. Overall, the field has shifted from direct model transfer towards designing specialized detectors that better balance accuracy, speed, and robustness across diverse underwater conditions.

**Multi-Modal Object Detection.** Multi-modal object detection (Chen et al. (2024b); Li et al. (2025)) integrates complementary sensing modalities—typically RGB for capturing color and texture, and infrared (IR) for contours and thermal cues—to enable robust perception under challenging conditions such as low light or haze. Public benchmarks such as FLIR (Zhang et al. (2020a)) and LLVIP (Jia et al. (2021)) have driven RGB-IR systems that extend single-stream backbones with parallel IR encoders and employ cross-modal or illumination-aware attention mechanisms (Guan et al. (2019); Li et al. (2019)) to prioritize the most informative modality. While these designs deliver sizable accuracy gains, they incur extra cost in sensors, power and on-board compute—constraints that are even harsher underwater. In the domain of underwater object detection, the pioneering AO-UOD model (Yu et al. (2025)) introduces a dual-stream backbone that facilitates feature exchange between optical images and sonar data, demonstrating the promise of acoustic-optical fusion. However, its dependence on a private paired dataset underscores the lack of a public benchmark, leaving multi-modal underwater object detection a largely underexplored frontier.

**Spiking Neural Network.** Spiking neural networks (SNNs) encode information as discrete spikes over time, offering a more biologically plausible alternative to ANNs. Early studies introduced temporal coding in SNNs, with SpikeProp (Bohte et al. (2000)) employing gradient-based learning and Tempotron (Gütig & Sompolinsky (2006)) proposing a biologically inspired rule for spike-time-based classification. Subsequent approaches either convert trained ANNs into SNNs (Bu et al. (2022); Qu et al. (2024)) or enable direct training through surrogate gradients (Neftci et al. (2019)) and three-factor learning rules (Frémaux & Gerstner (2016)). Recent work has integrated SNNs into vision backbones (such as Spiking-YOLO (Kim et al. (2020)), EMS-YOLO (Su et al. (2023))), and deployed them on neuromorphic hardware platforms like Intel Loihi and IBM TrueNorth. Due to their sparse firing and event-driven nature, SNNs transmit and process information only when and where neural activity occurs, achieving orders-of-magnitude reductions in computational cost and latency on dedicated hardware. These efficiencies make SNNs particularly well-suited for always-on, low-power applications in edge sensing and robotics.

## 3 METHOD

In this section, we first provide an overview of the proposed MMSNN framework, then describe the spike-driven multi-modal fusion module in detail, and finally analyze the computational efficiency advantages of our SNN compared to traditional ANNs.

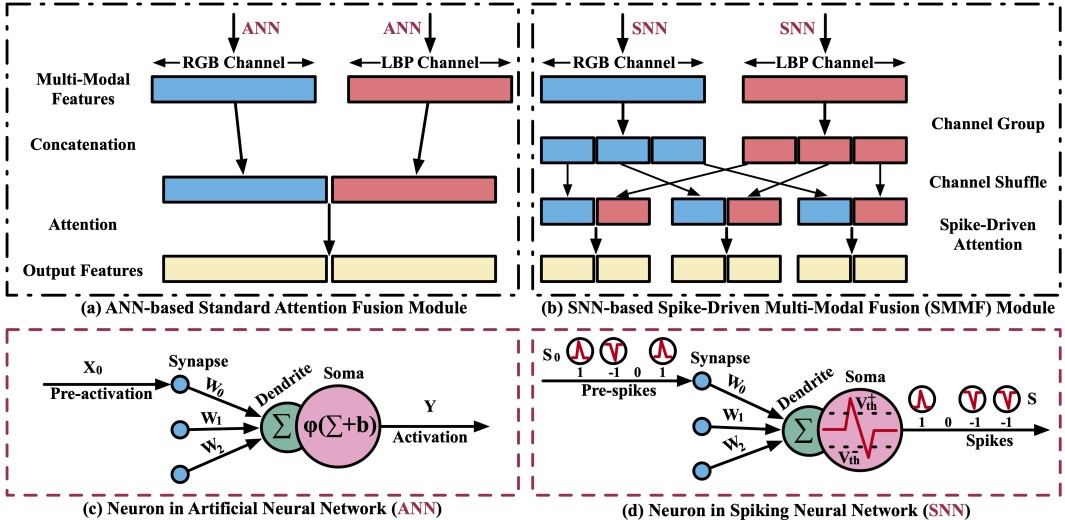

Figure 1: Comparison between (a) the ANN-based standard attention fusion (SAF) module and (b) the proposed spike-driven multi-modal fusion (SMMF) module. The SMMF module leverages a SNN architecture to reduce computational cost. Channel grouping and shuffling promote local cross-modal feature interaction and enhance feature diversity. The spike-driven attention mechanism introduces spiking neurons to lower computational consumption during dense attention computation.

### 3.1 OVERVIEW OF MMSNN ARCHITECTURE

The proposed MMSNN framework is adapted from the ANN-based YOLOX architecture (Ge et al. (2021)), extending its original single-branch design into a spike-based dual-branch structure that integrates both RGB and LBP modalities. Unlike ANN-based frameworks, it employs signed spiking neurons as the activation functions, which reduce computational cost and enhance compatibility with spiking neural networks. The LBP modality offers two main advantages over the commonly used infrared (IR) modality: (1) Richer texture representation–While IR captures object contours based on thermal signals and works well in darkness, LBP focuses on local intensity patterns, making it more effective for detecting textures under different lighting conditions. (2) Lower cost-IR requires specialized, expensive sensors, whereas LBP can be computed directly from standard RGB images without any extra hardware, making it a more cost-effective solution.

While multi-modal fusion typically increases computational complexity, we address this challenge by converting the ANN-based framework into a spike-based, computationally efficient SNN architecture. As illustrated in Fig. 1, RGB and LBP inputs are processed through two parallel SNN branches to extract deep RGB and LBP features, respectively. These features are then integrated by a spike-driven fusion module, which combines the complementary information into a unified representation. The fused features are subsequently forwarded to the detection head, which consists of three sub-heads: a classification head for object class prediction, a regression head for bounding-box localization, and an objectness head for distinguishing foreground objects from background regions. The overall objective function ($\mathcal{L}$) is composed of three components: Binary Cross Entropy (BCE) loss ($\mathcal{L}_{BCE}$) for classification, Intersection over Union (IoU) loss ($\mathcal{L}_{IoU}$) for bounding-box regression, and objectness loss ($\mathcal{L}_{obj}$) for suppressing irrelevant background regions:

$$\mathcal{L} = \mathcal{L}_{BCE} + \mathcal{L}_{IoU} + \mathcal{L}_{obj} \tag{1}$$

### 3.2 THE SPIKE-DRIVEN MULTI-MODAL FUSION MODULE

As illustrated in Fig. 1, conventional multi-modal fusion modules typically extract features from two modalities, denoted as $\mathbf{F}_{RGB} \in \mathbb{R}^{C \times H \times W}$ and $\mathbf{F}_{LBP} \in \mathbb{R}^{C \times H \times W}$, using separate ANN-based branches. These features are concatenated along the channel dimension to form $\mathbf{F}_{concat} = [\mathbf{F}_{RGB}; \mathbf{F}_{LBP}] \in \mathbb{R}^{2C \times H \times W}$, and passed through an attention module to enhance cross-modal interactions. However, this design presents two key limitations: (1) it increases computational costs

due to the parallel processing of dense, high-dimensional features and attention computations; (2) features from two modalities often reside far apart in feature space, making it difficult for the attention mechanism to learn effective long-range dependencies across modalities.

To overcome these challenges, we propose an efficient spike-driven module that incorporates channel grouping, channel shuffling, and a spike-driven attention mechanism. First, modality-specific features are extracted using lightweight SNN-based sub-networks, which inherently reduce computational cost. Let $\mathbf{F}RGB$ and $\mathbf{F}LBP$ denote the features extracted via the SNN branches. These features are divided into $G$ groups along the channel dimension:

$$\mathbf{F}_{RGB} = [\mathbf{F}_{RGB}^1, \ldots, \mathbf{F}_{RGB}^G], \quad \mathbf{F}_{LBP} = [\mathbf{F}_{LBP}^1, \ldots, \mathbf{F}_{LBP}^G] \tag{2}$$

Next, channel shuffling is applied to mix corresponding groups from both modalities, forming short-distance fused feature groups:

$$\mathbf{F}_{fused}^g = \text{Shuffle}([\mathbf{F}_{RGB}^g; \mathbf{F}_{LBP}^g]), \quad g = 1, \ldots, G \tag{3}$$

Each fused group $\mathbf{F}_{fused}^g$ is then passed through a spike-driven attention module $\mathcal{A}_{\text{spike}}$ to enhance modality interaction and emphasize salient features. The module $\mathcal{A}_{\text{spike}}$ is adopted from the Squeeze-and-Excitation block, with the ReLU activation replaced by signed spiking neurons to further reduce computation. The attention-enhanced output for group $g$ is given by:

$$\mathbf{F}_{attn}^g = \mathcal{A}_{\text{spike}}(\mathbf{F}_{fused}^g) \tag{4}$$

Finally, all attention-refined groups are concatenated to produce the final fused representation:

$$\mathbf{F}_{final} = [\mathbf{F}_{attn}^1, \ldots, \mathbf{F}_{attn}^G] \tag{5}$$

This localized fusion strategy not only improves efficiency but also enhances cross-modal interactions by operating in a short-distance feature space. Moreover, the use of grouping and shuffling reduces redundant and overlapping information within each modality, leading to more discriminative and diverse feature representations.

### 3.3 COMPARISON OF COMPUTATIONAL COST BETWEEN ANNS AND SNNS

**Computational Cost in Traditional ANNs.** As illustrated in Fig. 1, a standard ANN neuron performs the following computation:

$$y = \varphi\left(\sum_{i=1}^n w_i x_i + b\right) \tag{6}$$

where $x_i \in \mathbb{R}$ are continuous-valued inputs, $w_i \in \mathbb{R}$ are weights, $b \in \mathbb{R}$ is a bias term, $\varphi$ is a nonlinear activation function (e.g., ReLU), and $n$ is the number of inputs to the neuron.

According to Eq. 6, each neuron performs $n$ multiplications and $n$ additions ($n - 1$ additions for the weighted sum and one for adding the bias). Therefore, **the computational cost per neuron per timestep in an ANN** is approximately:

$$E_{\text{ANN}} \propto n \cdot E_{\text{mul}} + n \cdot E_{\text{add}} \tag{7}$$

where $E_{\text{mul}}$ and $E_{\text{add}}$ denote the computational costs of multiplication and addition, respectively.

**Computational Cost in SNNs with Signed Spiking Neurons.** In SNNs, neurons communicate via discrete spikes rather than continuous signals. Each neuron maintains a membrane potential $V(t)$. In SNNs, neurons communicate via discrete spikes rather than continuous signals. Each neuron maintains a membrane potential $V(t)$. A signed spike $S(t)$ is emitted when the membrane potential crosses a positive threshold $V_{\text{th}}^+$ or negative threshold $V_{\text{th}}^-$:

$$S(t) = \begin{cases} +1 & \text{if } V(t) \geq V_{\text{th}}^+ \\ -1 & \text{if } V(t) \leq V_{\text{th}}^- \\ 0 & \text{otherwise} \end{cases} \tag{8}$$

Here, $S(t) \in \{-1, 0, +1\}$ is the signed spike output at time $t$. The neuron remains inactive (i.e., no computation consumption) when $V(t) \in (V_{\text{th}}^{+}, V_{\text{th}}^{-})$. Only when $s_i(t) \neq 0$ does the postsynaptic neuron update its membrane potential:

$$V(t) = V(t-1) + \sum_{i=1}^{n} w_i \cdot s_i(t) \tag{9}$$

The operation in Eq. 9 eliminates the need for floating-point multiplications, reducing the computational cost to simple additions and subtractions, since the values of $s_i(t)$ are limited to $\pm 1$ or 0. Therefore, instead of full multiplications, only addition is required if $s_i(t) = +1$, subtraction if $s_i(t) = -1$, and no operation if $s_i(t) = 0$.

Let $\rho$ denote the average spike rate (typically $\rho \ll 1$, e.g., 0.1). Then, **the computational cost per neuron per timestep in an SNN** is approximately:

$$E_{\text{SNN}} \propto \rho \cdot n \cdot E_{\text{acc}} \tag{10}$$

where $E_{\text{acc}}$ is the computational cost for a single accumulation operation (addition or subtraction). In general, computation in SNNs is sparse, as it occurs only when spikes are generated, and it eliminates the need for expensive multiplications.

In ANNs, computation occurs at every timestep, leading to continuous computational consumption. In contrast, SNNs perform computations only when spikes are generated. The expected number of operations per neuron per timestep in an SNN is proportional to the average firing rate $\rho$, and since $\rho \ll 1$ (e.g., 0.1) in most practical SNNs, the total computation consumed per neuron per timestep is significantly lower than that of an ANN:

$$\rho \cdot n \cdot E_{\text{acc}} \ll n \cdot E_{\text{mul}} + n \cdot E_{\text{add}}, \qquad E_{\text{SNN}} \ll E_{\text{ANN}}. \tag{11}$$

This leads to a substantial reduction in computational cost compared to ANNs, primarily due to the sparsity of neuronal activity and the simplicity of the operations involved.

## 4 EXPERIMENTS

Table 1: The quantitative performance of representative detection frameworks on the RUOD dataset. The bold text represents the best performance, while the blue text indicates the second-best.

| | | Methods | Models | Backbones | Params | FLOPs | mAP | AP$_{0.50}$ | AP$_{0.75}$ | AP$_s$ | AP$_m$ | AP$_l$ |
|---|---|---|---|---|---|---|---|---|---|---|---|---|
| Non-Spiking | Generic | RepPoints | ResNet101 | 55.82M | 256.00G | 53.2 | 82.2 | 60.1 | 28.2 | 44.9 | 57.8 |
| | | FoveaBox | ResNet101 | 56.68M | 268.29G | 44.8 | 80.2 | 45.2 | 18.0 | 37.5 | 49.1 |
| | | ATSS | ResNet101 | 51.13M | 267.26G | 54.0 | 80.3 | 60.2 | 18.0 | 40.0 | 59.5 |
| | | DetectoRS | DResNet50 | 123.23M | 90.05G | 53.3 | 84.1 | 58.7 | 30.8 | 46.6 | 57.8 |
| | | YOLOv10 | CSPNet | 24.40M | 120.30G | 55.5 | 84.7 | 62.5 | 21.9 | 47.0 | 60.5 |
| | Undererwater | BoostRCNN | ResNet50 | 45.95M | 54.71G | 53.9 | 80.6 | 59.5 | 11.6 | 39.0 | 59.3 |
| | | RFTM | ResNet50 | 75.58M | 91.06G | 53.3 | 80.2 | 57.7 | 11.8 | 39.2 | 59.3 |
| | | ERLNet | SiEdgeR50 | 45.95M | 54.71G | 54.8 | 83.1 | 60.9 | 14.7 | 41.4 | 59.8 |
| | | GCCNet | SwinFT | 38.31M | 78.93G | 56.1 | 83.2 | 60.5 | 11.7 | 41.9 | 62.1 |
| | | DJLNet | ResNet50 | 58.48M | 69.51G | 57.5 | 83.7 | 62.5 | 15.5 | 41.8 | 63.1 |
| Spiking | SNN | Spiking-YOLO | TinyYOLO | 23.1M | 136.9G | 49.8 | 80.7 | 55.1 | 17.6 | 42.5 | 54.4 |
| | | EMS-YOLO | EMSResNet34 | 14.40M | 37.00G | 52.0 | 82.9 | 58.4 | 19.3 | 44.2 | 57.4 |
| | | SpikingYOLOX | SNNCSPNet | 49.53M | 151.69G | 57.0 | 84.2 | 61.2 | 11.0 | 41.2 | 63.0 |
| | Ours | MMSNN-T | MMSNN-T | 4.40M | 20.56G | 55.7 | 83.9 | 59.8 | 11.2 | 40.7 | 61.6 |
| | | MMSNN-S | MMSNN-S | 7.80M | 33.62G | 56.7 | 84.6 | 61.0 | 11.4 | 41.3 | 62.6 |
| | | MMSNN-L | MMSNN-L | 49.61M | 167.29G | 59.0 | 85.3 | 64.0 | 14.1 | 43.4 | 65.0 |

### 4.1 IMPLEMENTATION DETAILS

The MMSNN frameworks are implemented using PyTorch and SpikingJelly (Fang et al. (2023)), an open-source SNN library built on top of PyTorch. Following the YOLOX architecture (Ge et al.

Table 2: The quantitative performance of representative detection frameworks on the DUO dataset. The bold text represents the best performance, while the blue text indicates the second-best.

| Methods | | Models | Backbones | Params | FLOPs | mAP | $AP_{0.50}$ | $AP_{0.75}$ | $AP_s$ | $AP_m$ | $AP_l$ |
|---|---|---|---|---|---|---|---|---|---|---|---|
| Non-Spiking | Generic | RepPoints | ResNet101 | 55.82M | 256.00G | 59.4 | 80.4 | 70.1 | 55.5 | 59.6 | 60.1 |
| | | FoveaBox | ResNet101 | 55.24M | 286.72G | 53.7 | 78.4 | 63.9 | 55.3 | 54.3 | 54.6 |
| | | ATSS | ResNet101 | 51.11M | 286.72G | 55.4 | 79.2 | 63.2 | 55.7 | 55.7 | 56.0 |
| | | DetectoRS | DResNet50 | 123.23M | 90.05G | 58.9 | 81.4 | 68.3 | 49.6 | 57.6 | 61.8 |
| | | YOLOv10 | CSPNet | 24.40M | 120.30G | 62.3 | 84.7 | 70.9 | 48.5 | 63.9 | 61.6 |
| | Underewater | BoostRCNN | ResNet50 | 45.95M | 54.71G | 53.9 | 80.6 | 59.5 | 11.6 | 39.0 | 59.3 |
| | | RFTM | ResNet50 | 75.58M | 91.06G | 60.1 | 79.4 | 68.1 | 49.0 | 61.1 | 59.5 |
| | | ERLNet | SiEdgeR50 | 45.95M | 54.71G | 61.2 | 81.4 | 69.5 | 55.2 | 62.2 | 60.8 |
| | | GCCNet | SwinFT | 38.31M | 78.93G | 61.1 | 81.6 | 67.3 | 52.5 | 63.6 | 59.3 |
| | | DJLNet | ResNet50 | 58.48M | 69.51G | 65.6 | 84.2 | 73.0 | 55.6 | 67.4 | 64.1 |
| Spiking | SNN | Spiking-YOLO | TinyYOLO | 23.1M | 136.9G | 60.6 | 78.8 | 67.3 | 52.0 | 61.2 | 59.3 |
| | | EMS-YOLO | EMSResNet34 | 14.40M | 37.00G | 62.7 | 80.8 | 69.1 | 53.8 | 63.5 | 61.1 |
| | | SpikingYOLOX | SNNCSPNet | 49.53M | 151.69G | 64.3 | 81.9 | 70.4 | 54.7 | 65.0 | 63.5 |
| | Ours | MMSNN-T | MMSNN-T | **4.40M** | 20.56G | 65.1 | 83.8 | 72.0 | 55.6 | 66.0 | 64.0 |
| | | MMSNN-S | MMSNN-S | 7.80M | 33.62G | 66.5 | 85.1 | 73.2 | **56.5** | **68.0** | 65.3 |
| | | MMSNN-L | MMSNN-L | 49.61M | 167.29G | **67.1** | **86.3** | **73.5** | 56.2 | 68.9 | **66.6** |

(2021)), we design model variants at different scales: Tiny (**MMSNN-T**), Small (**MMSNN-S**), and Large (**MMSNN-L**). The model is trained using the Adam optimizer with a StepLR scheduler, an initial learning rate of 0.01, and a batch size of 16. The total number of training epochs are adjusted based on the model size: 500 epochs for MMSNN-L, and 300 epochs for both MMSNN-T and MMSNN-S. All the experiments are conducted on a server with an Intel(R) Xeon(R) Silver 4114 CPU @ 2.20GHz and a single Tesla V100 GPU with a 32GB memory.

### 4.2 EVALUATION DATASETS AND METRICS

**Datasets.** We evaluate the proposed MMSNN framework on the RUOD (Fu et al. (2023b)) and DUO (Liu et al. (2021)) datasets. RUOD includes 9,800 training images and 4,200 testing images across ten underwater object categories, covering diverse object types and challenging visual conditions. DUO consists of 6,671 training images and 1,111 testing images.

**Evaluation Metrics.** Model performance is assessed using the COCO evaluation metrics, including the mean Average Precision (mAP) across a range of IoU thresholds ($mAP@[0.5:0.05:0.95]$), as well as average precision at fixed IoU thresholds ($AP_{0.50}$ and $AP_{0.75}$). To evaluate detection performance across different object sizes, we report Average Precision for small ($AP_S$), medium ($AP_M$), and large ($AP_L$) objects. In addition to detection accuracy, we assess model efficiency by reporting the computational cost in FLOPs (floating point operations) and model complexity in terms of parameter count (Params).

### 4.3 COMPARISONS WITH STATE-OF-THE-ART METHODS

We compare our proposed MMSNN models against a range of state-of-the-art object detectors, including five generic detectors (YOLOv10 (Wang et al. (2024a)), RepPoints (Yang et al. (2019)), FoveaBox (Kong et al. (2020)), ATSS (Zhang et al. (2020d)), and DetectoRS (Qiao et al. (2021)), five top-performing underwater detectors (DJLNet (Wang et al. (2024b), GCCNet (Dai et al. (2024), ERLNet (Dai et al. (2023), RFTM (Fu et al. (2023a), and BoostRCNN (Song et al. (2023)), and three leading SNN-based detectors (SpikingYOLOX (Miao et al. (2025), EMS-YOLO (Su et al. (2023), and Spiking-YOLO (Kim et al. (2020)).

**Precision Analysis:** As shown in Tables 1 and 2, the proposed MMSNN-L achieves state-of-the-art detection performance. DJLNet ranks second with 57.5% mAP, benefiting from its tailored design for underwater object detection. It fuses appearance and edge features to enhance robustness in complex environments, while its image decolorization module corrects color distortions caused by light

Table 3: The performance comparison between MMSNNs variants with (*w/*) and without (*w/o*) the Multi-Modal Architecture (MMA), as well as between models using the standard attention-based fusion (SAF) module and those using the Spike-Driven Multi-Modal Fusion (SMMF) module.

| Models | MMSNN-T | | MMSNN-S | | MMSNN-L | |
|---|---|---|---|---|---|---|
| **MMA** | *w/o* MMA | *w/* MMA | *w/o* MMA | *w/* MMA | *w/o* MMA | *w/* MMA |
| **mAP** | 53.2 | 55.7+2.5% | 54.9 | 56.7+1.8% | 57.6 | 59.0+1.4% |
| **AP$_{0.50}$** | 82.5 | 83.9+1.4% | 83.6 | 84.6+1.0% | 84.7 | 85.3+0.6% |
| **AP$_{0.75}$** | 57.5 | 59.8+2.3% | 58.3 | 61.0+2.7% | 62.0 | 64.0+2.0% |
| **SMMF** | *w/* SAF | *w/* SMMF | *w/* SAF | *w/* SMMF | *w/* SAF | *w/* SMMF |
| **mAP** | 54.3 | 55.7+1.4% | 55.5 | 56.7+1.2% | 58.3 | 59.0+0.7% |
| **AP$_{0.50}$** | 83.1 | 83.9+0.8% | 84.2 | 84.6+0.4% | 85.0 | 85.3+0.3% |
| **AP$_{0.75}$** | 58.6 | 59.8+1.2% | 59.4 | 61.0+1.6% | 63.1 | 64.0+0.9% |

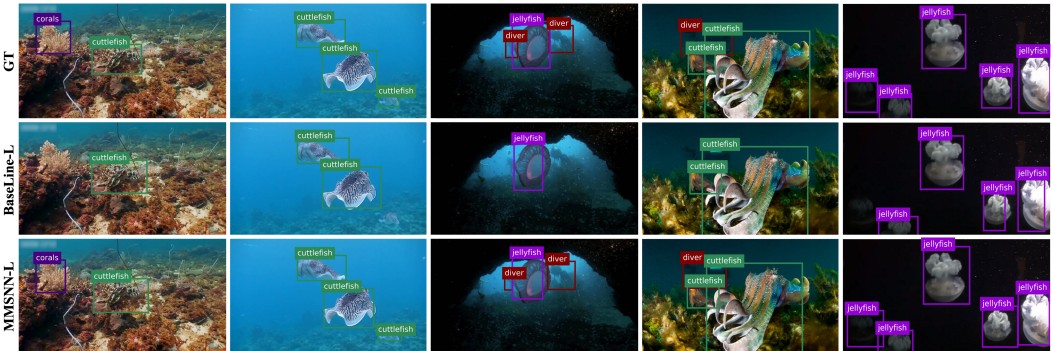

Figure 2: Visualization of detection results comparing the proposed MMSNN-L model (with both RGB and LBP modalities) to the baseline model (using only the RGB modality). The baseline frequently fails to detect objects in low-light conditions, whereas MMSNN consistently achieves accurate detection under varying illumination.

absorption and its edge enhancement branch refines object boundary localization. SpikingYOLOX, a general-purpose SNN detector, ranks third, highlighting the potential of SNNs for underwater object detection and motivating our adaptation of SNN for this domain. In contrast, MMSNN-L leverages a multi-modal architecture that integrates complementary information from RGB and LBP features, resulting in a significant improvement in detection accuracy.

**Efficiency Analysis:** Tables 1 and 2 also compare model efficiency using FLOPs and Params. On RUOD, While our MMSNN-S ranks fourth in terms of mAP (56.7%), slightly behind DJNet (57.5%) and SpikingYOLOX (57.0%), it offers significantly higher efficiency. Specifically, MMSNN-S requires substantially fewer parameters (7.80M vs 58.48M vs 49.53M) and FLOPs (33.62G vs 69.51G vs 151.69G) compared to the other two models. These results demonstrates that MMSNN-S strikes an excellent balance between detection performance and computational cost, offering comparable precision with markedly lower resource consumption.

### 4.4 ABLATION STUDY

To validate the effectiveness of the proposed MMSNN frameworks, we perform an ablation study on the RUOD dataset, focusing on three key components: the Multi-Modal Architecture (MMA), the Spike-Driven Multi-Modal Fusion (SMMF) module, and the use of signed spiking neurons.

**Effectiveness of the Multi-Modal Architecture (MMA):** We compare the full multi-modal architecture–which includes both RGB and LBP branches–with a single-modal **baseline** that uses only the RGB branch. As shown in Table 3, the MMSNN models with MMA consistently outperform their single-modal counterparts across all three model variants. This performance gain is attributed to the complementary strengths of the RGB and LBP modalities: RGB images capture rich

Table 4: The performance comparison between MMSNNs with IF and signed spiking neurons.

| Models | MMSNN-T | | MMSNN-S | | MMSNN-L | |
|---|---|---|---|---|---|---|
| Neurons | IF | Signed | IF | Signed | IF | Signed |
| mAP | 54.5 | 55.7+1.2% | 55.6 | 56.7+1.1% | 57.8 | 59.0+1.2% |
| $AP_{0.50}$ | 83.1 | 83.9+0.8% | 83.9 | 84.6+0.7% | 84.7 | 85.3+0.6% |
| $AP_{0.75}$ | 58.6 | 59.8+1.2% | 59.3 | 61.0+1.7% | 62.1 | 64.0+1.9% |

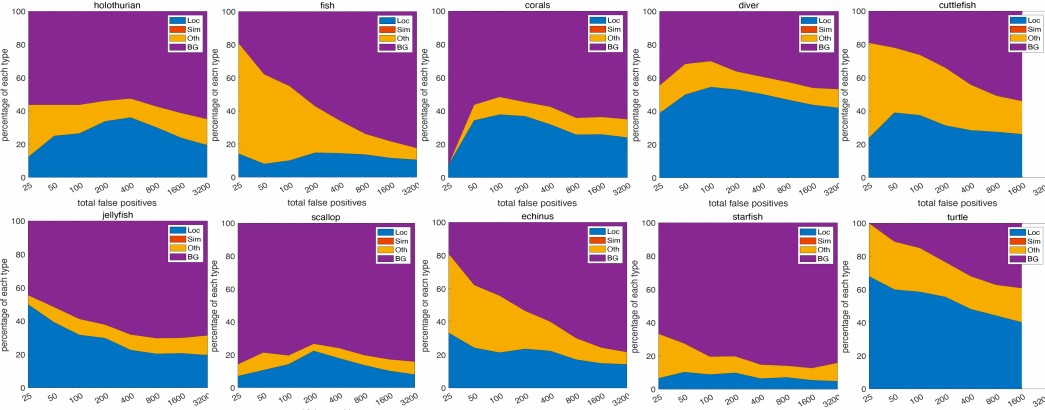

Figure 3: The distribution of error types made by MMSNN-L for each category on the RUOD dataset. The error types include localization error (Loc), confusion with similar categories (Sim), confusion with other dissimilar categories (Oth), and misclassifications as background (BG).

visual and color information, while LBP features offer robust local texture and edge representations, especially under varying illumination conditions.

Illumination variability significantly affects detection performance. As illustrated in Fig. 2, the baseline model without MMA frequently fails to detect objects under low-light scenarios. In contrast, the MMSNN-L model with MMA maintains reliable detection across diverse lighting environments. This robustness is primarily attributed to the LBP modality, which provides illumination-invariant features, thereby enhancing the model's resilience to lighting changes.

**Effectiveness of the Spike-Driven Multi-Modal Fusion (SMMF) Module:** We evaluate the proposed SMMF module (Fig. 1 (b)) against a standard attention-based fusion (SAF) module (Fig. 1 (a)). The SAF module fuses deep RGB and LBP features via simple concatenation, followed by an ANN-based Squeeze-and-Excitation (SE) attention block. In contrast, the SMMF module integrates three key components: channel grouping, channel shuffling, and a spike-driven attention mechanism. This attention module is adapted from the SE block by replacing ReLU activations with signed spiking neurons, ensuring full compatibility with spiking neural networks.

Table 3 presents performance comparisons between the two fusion strategies across the MMSNN-T, MMSNN-S, and MMSNN-L variants. In all cases, models equipped with the SMMF module outperform those using SAF, demonstrating the effectiveness of the proposed design. The performance gains are primarily attributed to the channel grouping and shuffling strategies, which facilitate local cross-modal feature interaction, reduce redundancy, and promote feature diversity—ultimately enhancing the model's representational capacity and detection performance.

**Effectiveness of Signed Spiking Neurons:** In this study, we evaluate two widely used spiking neurons for ANN-to-SNN conversion: the Integrate-and-Fire (IF) neuron Jin et al. (2023) and the signed spiking neuron Kim et al. (2020). Table 4 compares MMSNN frameworks using each neuron type. The results consistently indicate that the models with signed spiking neurons outperform those using IF neurons. This performance gap primarily arises from the limitations of binary spike encoding in IF neurons, which can only represent the presence (1) or absence (0) of a signal. While such binary encoding is adequate for event-based data, it lacks the capacity to capture the richer, more complex feature representations required for visual data. In contrast, signed spiking neurons

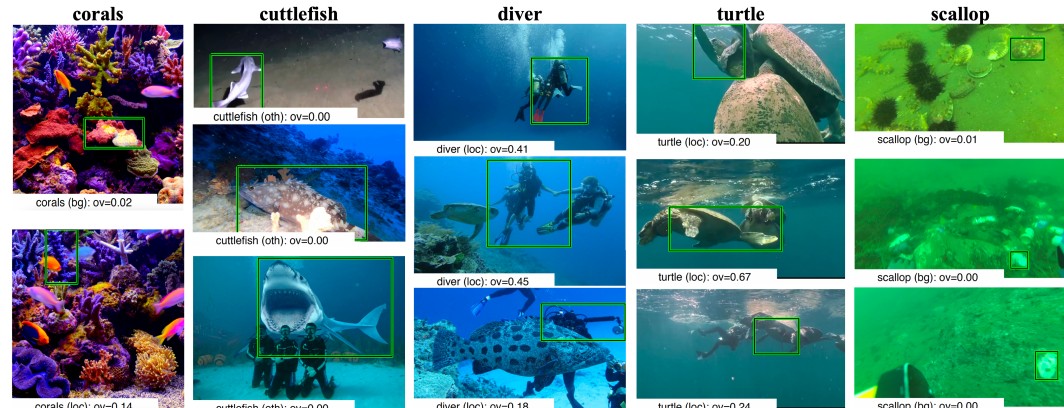

Figure 4: Visualization of false positives from MMSNN-T. Text labels indicate the error types: "loc" (localization error), "bg" (background confusion), and "oth" (confusion with other categories). "ov" represents the overlap ratio between the detected object and its ground truth.

introduce both positive and negative spikes (+1, –1), enabling bidirectional signaling. This enhanced encoding capacity allows for more nuanced and discriminative feature representation, leading to improved model expressiveness and better overall performance.

### 4.5 ERROR ANALYSIS OF MMSNN

To better understand the limitations of MMSNN and guide future improvements, we conduct a detailed error analysis of MMSNN-L using the detection analysis tool Diagnosis Hoiem et al. (2012). Errors are categorized into four types: localization errors (Loc), confusion with similar categories (Sim), confusion with dissimilar categories (Oth), and misclassifications as background (BG). For this analysis, all ten categories in the RUOD dataset are treated as dissimilar classes.

As shown in Fig. 3, different object categories tend to exhibit different dominant error types. For instance, scallop and starfish categories show a high incidence of background errors, where background regions are mistakenly identified as objects. This is largely due to the blurry and color-distorted visual characteristics of underwater imagery, which reduce the contrast between objects between objects and their surroundings. Fig. 4 further illustrates that stones are frequently misclassified as scallops due to their similar textures and colors under such conditions. Localization errors are another major source of inaccuracy, particularly for categories such as diver and turtle, where object occlusion or dense clustering make precise bounding box prediction more difficult. Additionally, inter-class confusion is common between visually similar categories such as cuttlefish and fish, which share overlapping appearance features. These observations underscore the need to address both external visual challenges (e.g., blur, color distortion) and intrinsic category-level similarities to further enhance detection accuracy in underwater environments.

## 5 CONCLUSIONS

In this paper, we presented MMSNN, a Multi-Modal Spiking Neural Network designed for computational efficient underwater object detection. By fusing RGB and LBP modalities within a spike-driven architecture, MMSNN leverages both visual richness and texture robustness while maintaining low computational overhead. Our fusion module—incorporating channel grouping, channel shuffling, and spike-driven attention—enables efficient and expressive multi-modal integration without the complexity of traditional dense ANN-based approaches. Experiments on the RUOD and DUO datasets show that MMSNN achieves strong detection performance while using less computation compared to traditional ANN-based methods. In future work, we plan to extend MMSNN to include additional data modalities, such as sonar and thermal imagery, to make it applicable to a wider range of underwater and low-visibility tasks.

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

# A APPENDIX

## A.1 USE OF LLMS

Large Language Models (LLMs) were used solely to assist with writing and polishing the text.

## A.2 CODE OF ETHICS AND ETHICS STATEMENT

The research conducted in the paper conform, in every respect, with the ICLR Code of Ethics https://iclr.cc/public/CodeOfEthics.

## A.3 REPRODUCIBILITY

This paper provides all necessary details to enable reproduction of the main experimental results, including dataset descriptions, training procedures, training parameters, and evaluation protocols.

