# OpenReview forum: "Multi-Modal Spiking Neural Network for Efficient and Robust Underwater Object Detection"
_ICLR.cc/2026/Conference — ICLR 2026 Conference Withdrawn Submission_

### Official Review · Reviewer_1zhM · 2025-10-28

**Soundness:** 2
**Presentation:** 2
**Contribution:** 2
**Rating:** 2
**Confidence:** 4

**Summary:**

This paper proposes MMSNN, a Multi-Modal Spiking Neural Network that fuses RGB and Local Binary Pattern (LBP) modalities for underwater object detection. The method introduces a Spike-Driven Multi-Modal Fusion (SMMF) module, which incorporates channel grouping, shuffling, and spiking attention. Experiments on two underwater datasets (RUOD and DUO) claim state-of-the-art performance while maintaining computational efficiency.

**Strengths:**

The paper is well-structured, and the proposed method is easy to follow. Experiments on two underwater datasets (RUOD and DUO) claim state-of-the-art performance while maintaining computational efficiency.

**Weaknesses:**

1. Limited Novelty: The approach mainly combines existing concepts—spiking neural networks (SNNs), LBP features, and channel grouping/shuffling—without introducing a fundamentally new algorithm or learning principle.
2. Unconvincing Motivation for Multimodality: The rationale for using multimodal inputs is weak. LBP is a traditional handcrafted texture descriptor derived from the RGB image, meaning the model does not learn cross-modal relationships in the conventional sense. No comparisons are provided with other texture-based augmentations or standard feature fusion methods to justify that the LBP channel adds meaningful diversity. Furthermore, the paper lacks comparisons with other state-of-the-art multimodal approaches, both in methodology and experiments.
3. Insufficient SNN Innovation: The proposed SNN is only compared against a standard ANN architecture. Comparisons with state-of-the-art SNN-based methods, such as SpikingYOLO, SpikeYOLO, and EMS YOLO, are missing. It is also unclear whether the proposed multimodal fusion provides any performance improvement over these established SNN approaches. Moreover, the method relies on PyTorch/SpikingJelly simulations rather than true event-driven computation, which undermines the claimed efficiency benefits.
4. Experimental Weaknesses: The reported accuracy gains are marginal (1–2% over baselines) and could easily result from architectural or training differences rather than the proposed mechanism. No statistical tests or confidence intervals are provided to assess significance. The datasets (RUOD, DUO) are small and single-domain (RGB), limiting the ability to convincingly demonstrate the model’s multimodal capabilities.

**Questions:**

Could the authors clarify the primary novelty of MMSNN beyond combining existing techniques, and justify the choice of LBP as a complementary modality with comparisons to other texture-based or multimodal methods?

How does the proposed SNN perform compared to state-of-the-art SNN-based detectors such as SpikingYOLO, SpikeYOLO, or EMS YOLO, and does multimodal fusion provide measurable improvements over these approaches?

Can the authors provide evidence of statistical significance for the reported gains and discuss potential deployment on event-driven hardware to support the claimed efficiency benefits?

---

### Official Review · Reviewer_UGws · 2025-10-29

**Soundness:** 2
**Presentation:** 2
**Contribution:** 1
**Rating:** 4
**Confidence:** 4

**Summary:**

This paper proposes an SNN-based underwater object detection network that leverages the low-power characteristics of SNNs to achieve compatibility between edge computing devices and performance, achieving promising results on two typical underwater object detection datasets. While the paper is well-written with clear logic and sufficient experiments, I find it challenging to identify the specific technical innovations that distinguish this work from prior art. I would appreciate further clarification from the authors on the novelty aspects, and I am open to raising my score if my concerns can be adequately addressed.

**Strengths:**

(1) The paper is well-organized with clear logic and fluent presentation.

(2) The figures and tables are well-designed and effectively convey the information.

(3) The experimental evaluation is comprehensive and thorough.

**Weaknesses:**

(1) The paper would benefit from an overall framework diagram, particularly a schematic illustration showing the specific improvements made to the YOLOX architecture.

(2) The position where SMMF modules are integrated into the framework could be clarified. Additionally, the term "MMA" appears in the tables but its definition is not immediately clear.

(3) While the paper constructs a multi-modal framework using LBP features, the motivation and theoretical justification for this design choice could be elaborated more thoroughly. It would be helpful to distinguish between methodological contributions and engineering applications.

(4) Given that power efficiency is a well-established characteristic of SNNs in the community, it might be more compelling to focus on other unique contributions or provide empirical power measurements to substantiate these claims.

(5) To better support the claims about power consumption advantages and edge deployment suitability, it would strengthen the paper to include actual deployment tests on edge devices and provide more detailed analysis of detection efficiency metrics.

**Questions:**

(1) Could the authors please clarify the core methodological innovations of this work? While adapting SNNs to underwater detection is valuable, it would be helpful to understand what novel techniques or insights are introduced beyond applying existing SNN frameworks to this domain.

(2) The SMMF module and its attention mechanism appear to share similarities with standard ANN components. Could the authors elaborate on what specific adaptations or innovations were made to tailor these components for SNNs?

(3) Regarding the attention mechanism, beyond replacing the activation function with spiking neurons, what other design innovations differentiate it from the standard SEBlock?

(4) To better understand the source of performance improvements, could the authors provide additional ablation studies comparing ANN-based models with LBP features versus the proposed SNN-based approach? This would help isolate the contribution of the SNN architecture itself from the benefits of introducing LBP features.

(5) The ablation studies could be expanded to include more detailed analysis of architectural design choices, such as the impact of SMMF placement at different layers, the effect of various hyperparameters, etc.

(6) To improve reproducibility, it would be helpful to include a complete framework flowchart showing the end-to-end pipeline and provide more detailed specifications of the model architecture and implementation details.

---

### Official Review · Reviewer_CKu9 · 2025-10-31

**Soundness:** 2
**Presentation:** 2
**Contribution:** 2
**Rating:** 2
**Confidence:** 5

**Summary:**

This paper proposes MMSNN, a Multi-Modal Spiking Neural Network for underwater object detection that fuses RGB and LBP features through a spike-driven architecture. The core contribution is the Spike-Driven Multi-Modal Fusion (SMMF) module using channel grouping, shuffling, and spike-driven attention. MMSNN-L achieves state-of-the-art performance with 59.0% mAP on RUOD and 67.1% mAP on DUO datasets while maintaining low computational cost.

**Strengths:**

1. Strong detection accuracy: MMSNN-L achieves state-of-the-art performance on both datasets, outperforming previous best methods including DJLNet and SpikingYOLOX.
2. Excellent computational efficiency: The model demonstrates superior efficiency with significantly fewer parameters and FLOPs while maintaining competitive accuracy.

**Weaknesses:**

1. This paper seems to use SNNs just for the sake of using SNNs. What specific problem do SNNs solve in this domain? Or what issues arise from applying SNNs to this field? I don't see the significance here.
2. Why use LBP as the second modality? How can it replace these other hardware components? I don't see the rationale.
3. The paper structure is quite unreasonable - the introduction only has 3 paragraphs.
4. The experimental comparisons are very unfair. The paper only compares against outdated methods, such as EMS-YOLO from 2023. As far as I know, this year's SpikingYOLOX is not a pure SNNs method.

**Questions:**

1. See the Weaknesses part.

---

### Official Review · Reviewer_nfqG · 2025-11-01

**Soundness:** 3
**Presentation:** 3
**Contribution:** 2
**Rating:** 4
**Confidence:** 4

**Summary:**

The paper proposes MMSNN, a Multi-Modal Spiking Neural Network for efficient underwater object detection. It combines standard RGB images with Local Binary Pattern (LBP) features to capture both visual and texture information while remaining illumination-robust. The key contribution is the Spike-Driven Multi-Modal Fusion (SMMF) module, which integrates channel grouping, channel shuffling, and spike-based attention to enable efficient cross-modal interaction within a spiking framework. The work positions SNNs as a potential low-power alternative for underwater and edge-sensing applications.

**Strengths:**

The paper offers a well-executed integration of spiking neural networks with multi-modal fusion for underwater object detection. Using LBP as a secondary modality derived from RGB is a clever and practical idea that improves robustness without extra sensors. The work is clearly written, methodically presented, and supported by consistent experiments and ablations. While the gains are moderate, the approach is original in combining spike-driven computation with texture-based fusion for a domain where efficiency and illumination invariance matter. Overall, it’s a technically sound and clearly communicated contribution with practical relevance.

**Weaknesses:**

1.  The architecture builds on existing parts — YOLOX backbone, LBP features, and squeeze-excitation style attention. The fusion block is mostly a spiking version of known grouping and attention ideas. The paper would be stronger if it showed how spike timing or temporal coding adds something new beyond replacing activations.

2.    The comparisons miss modern detectors like YOLOv8/9, RT-DETR, EfficientDet, or recent Transformer-based underwater models. Including these would show whether the gain comes from the SNN design or just older baselines.

3.    If the goal is only lower computation, pruning, quantization, or distillation could do that without moving to SNNs. The paper should explain why spikes are a better trade-off.

4. The mAP improvements are modest (+1–2). Training and tuning an SNN is harder than a CNN, so the cost–benefit is unclear. Reporting training time or stability would help.

**Questions:**

1.  Can the authors provide actual energy or latency results, ideally on neuromorphic or low-power hardware, to support the claim of computational efficiency? FLOPs alone are not sufficient for SNN evaluation.

2. Why were recent detectors such as YOLOv8/9, RT-DETR, or EfficientDet not included in the comparison? Were these models tested and found unsuitable, or omitted for resource reasons?

3.  How does training time and stability of MMSNN compare to standard CNN-based detectors of similar size? Is the surrogate-gradient optimization stable across runs?

4.   Beyond the reduced multiplications, do the spiking activations or temporal dynamics contribute any measurable representational benefit? For example, lower spike rate, faster convergence, or improved robustness under noise?

5. Since LBP is derived from the same RGB image, can the authors clarify how much additional information it provides versus simply using standard multi-scale RGB features or learned texture embeddings?

---

### Note · Authors · 2025-11-12

**Comment:**

Apply for Withdrawal

**Withdrawal Confirmation:**

I have read and agree with the venue's withdrawal policy on behalf of myself and my co-authors.